# Physical Activity Evaluation Using Activity Trackers for Type 2 Diabetes Prevention in Patients with Prediabetes

**DOI:** 10.3390/ijerph19148251

**Published:** 2022-07-06

**Authors:** Antanas Bliudzius, Kristina Svaikeviciene, Roma Puronaite, Vytautas Kasiulevicius

**Affiliations:** 1Family Medicine and Oncology, Clinic of Internal Diseases, Faculty of Medicine, Vilnius University, M.K. Ciurlionio Str. 21/27, LT-03101 Vilnius, Lithuania; kristina.svaikeviciene@mf.vu.lt (K.S.); vytautas.kasiulevicius@mf.vu.lt (V.K.); 2Institute of Data Science and Digital Technologies, Vilnius University, Akademijos Str. 4, LT-08412 Vilnius, Lithuania; roma.puronaite@santa.lt

**Keywords:** prediabetes, activity trackers, variability

## Abstract

Background: Prediabetes is a reversible condition, but lifestyle-changing measures, such as increasing physical activity, should be taken. This article explores the use of Fitbit activity trackers to assess physical activity and its impact on prediabetic patient health. Methods: Intervention study. In total, 30 volunteers (9 males and 21 females), aged 32–65 years, with impaired glucose levels and without diabetes or moving disorders, received Fitbit Inspire activity trackers and physical activity recommendations. A routine blood check was taken during the first and second visits, and body composition was analyzed. Physical activity variability in time was assessed using a Poincare plot. Results: The count of steps per day and variability differed between patients and during the research period, but the change in total physical activity was not statistically significant. Significant positive correlations between changes in lipid values, body mass composition, and variability of steps count, distance, and minutes of very active physical activity were observed. Conclusions: When assessing physical activity, data doctors should evaluate not just the totals or the medians of the steps count, but also physical activity variability in time. The study shows that most changes were better linked to the physical activity variability than the total count of physical activity.

## 1. Introduction

Prediabetes is a term used increasingly to describe people with impaired glucose tolerance (IGT) and/or impaired fasting glucose (IFG) [1]. Impaired glucose tolerance (IGT) and impaired fasting glucose (IFG) are conditions of raised blood glucose levels above the normal range and below the diabetes diagnostic threshold. The importance of IGT and IFG is three-fold: first, they signify a higher risk of the future development of type 2 diabetes mellitus (T2D); second, IGT and IFG indicate an already heightened risk of cardiovascular disease; and third, their detection opens the door to interventions that can lead to the prevention of type 2 diabetes [2,3,4,5,6]. In 2021, 541 million adults, or 10.6% of adults worldwide, were estimated to have IGT. By 2045, this figure is projected to increase to 730 million adults, or 11.4% of all adults. In 2021, there were an estimated 319 million adults, or 6.2% of the global adult population with IFG. An estimated 441 million adults, or 6.9% of the global adult population, are projected to have IFG in 2045 [1]. This makes prediabetes one of the most rapidly growing health problems. The most common causes of this pathology are overweight and obesity and the main reasons for this are malnutrition and low physical activity [7]. Modification in lifestyle plays an important role in avoiding the prognosis of type 2 diabetes (T2D) and its complications in the future. One such lifestyle change that has been positively correlated with stopping the progression of T2D in those with prediabetes is increasing physical activity (PA) [8].

Just 150 min/week of moderate-intensity physical activity, such as brisk walking, showed beneficial effects in those with prediabetes [9]. Several major randomized controlled trials, including the Diabetes Prevention Program (DPP) [9], the Finnish Diabetes Prevention Study (DPS) [10], and the Da Qing Diabetes Prevention Study (Da Qing study) [11], demonstrate that lifestyle/behavioral therapy with an individualized reduced-calorie meal plan is highly effective in preventing or delaying type 2 diabetes and improving other cardiometabolic markers (such as blood pressure, lipids, and inflammation) [12]. The most substantial evidence for diabetes prevention in the U.S. comes from the DPP trial [9]. The DPP demonstrated that intensive lifestyle intervention could reduce the risk of incident type 2 diabetes by 58% over three years. 

One of first steps in improving healthy lifestyle behavior is physical activity evaluation. Tools used to assess physical activity (PA) are questionnaires or physical activity diaries and objective data collection. Objective data collections can be divided into four groups. These are the measurement of energy consumption, the measurement of physiological parameters, the measurement of movements, and positioning [13]. Currently, the most popular method for PA estimation is direct motion measurement. Pedometers and accelerometers are used for this purpose. In recent years, we went through enormous mobile devices breakthroughs, and now we can count our steps using smartphones, smartwatches, and activity trackers. Smart bracelets measure the daily number of steps with sufficient accuracy and are suitable for long-term physical activity assessment [14,15,16,17]. 

At present, diabetes and cardiovascular diseases in Lithuania are among the leading causes of illness and death [18]. The country does not have a unified obesity and diabetes prevention program at the state level, but the greatest prevention is done by family doctors’ practice. The increasing popularity of physical activity trackers gives us the opportunity to evaluate physical activity more objectively and use it in practice. Studies reveal that the use of modern technologies helps not only to evaluate PA but also to improve it [19]. That leads to a reduction of body fat, cholesterol, and glycated hemoglobin levels [20,21]. 

Our aims were to evaluate the use of Fitbit activity trackers in the assessment of physical activity and its variability in time, and to evaluate physical activity’s impact on body fat, glycated hemoglobin, and cholesterol levels in patients with prediabetes. We used Fitbit activity trackers as the most common tools to reach this goal and evaluate their benefits and limitations [22].

## 2. Materials and Methods

### 2.1. Recruitment

This intervention study evaluated Fitbit data clinical use opportunities in primary care and diabetes prevention. Eligible participants, aged from 18 to 65 years old, had increased fasting glucose levels (5.6 to 6.9 mmol/L), had no health issues with the skeletal or muscular system, and other conditions that could compromise their ability to move [23]. Inclusion criteria consisted of voluntary participation in the study and had to have a smartphone with the latest operating system (Apple iOS 13, Android OS 8.0 or later), email address, and being able to use it.

The thirty volunteers were visiting family doctors at Vilnius University Hospital Santaros Klinikos (VUHSK) Family medicine center. Participants received a Fitbit Inspire activity tracker (AT), and they allowed researchers to collect data from their accounts after six months of wearing it.

There were two visits to the Family medicine center. During the first visit, we performed body mass composition analysis with the medical bioimpedance device X-contact 356 and made a routine blood test. Total cholesterol, triglycerides, low-density lipoproteins, high-density lipoproteins, and glycated hemoglobin were performed. After all the tests, the patients were consulted by their family doctor, and recommendations on a healthy diet and physical activity (based on World Health Organisation recommendations) were given.

The second visit was after six months. We repeated the same procedures as in the first visit, and after all, we asked participants to log in to their accounts at accounts.fitbit.com/login and exported data from the research period.

The study was approved by the Vilnius Regional Biomedical Research Ethics Committee (approval no. 2019/6-1143-634). Fitbit was not involved in the design, implementation, data analysis, or manuscript preparation of the study.

### 2.2. Intervention Tools

Every participant was given a Fitbit Inspire activity tracker. Fitbit is a consumer product to monitor physical activity such as steps, distance, physical activity levels, and sedentary time. These are commonly small devices worn on the wrist and can be used individually by the screen on the device or connected by a smartphone and mobile application. The Fitbit activity tracker uses a microelectronic triaxial accelerometer to capture body motion in 3 dimensions, and then these motions are analyzed by the alghorytmes and converted to physical activity data. These consumer devices showed respective data accuracy and are more commonly used in clinical trials [24]. Every participant was taught how to use these tools and was given full technical support during the study.

### 2.3. Data Analysis

The data obtained from the smart bracelets were: physical activity recording data, steps per day, distance, minutes sedentary, minutes weakly active, active, minutes fairly active, and minutes very active [25,26].

We adapted the Poincaré plot mathematical model to evaluate physical activity variability. This method is widely used to study physiological signals [27,28]. To analyze physical activity variability, the Poincaré plot method with a time delay of one day (i = 1) was applied and additional measures for selected physical activity time series accessed from patients’ Fitbit data were calculated. An average value (AVG), standard deviation (SD), and some standard Poincaré plot parameters representing short-term variability (SD1), long-term variability (SD2), the ratio of SD1 and SD2 (SD12), and area of fitting ellipse (AFE) were selected to analyze the variability.
$$\begin{array}{l} \mathrm{SD}1=\frac{\surd 2}{2}*\mathrm{SD}\left(x_{n}-x_{n+1}\right)\\ \mathrm{SD}2=\sqrt{2\mathrm{SD}{\left(x_{n}\right)}^{2}-\frac{1}{2}\mathrm{SD}{\left(x_{n}-x_{n+1}\right)}^{2}}\\ \mathrm{SD}12=\frac{\mathrm{SD}1}{\mathrm{SD}2}\\ \mathrm{AFE}=\pi *\mathrm{SD}1*\mathrm{SD}2 \end{array}$$

### 2.4. Statistical Analysis

The Shapiro–Wilk test was used for normality testing. Paired *t*-test or Wilcoxon’s matched-pairs signed-rank test were used for baseline and follow-up data comparison. Pearson’s or Spearman’s correlation was performed to assess the association between changes in selected variables and physical activity variability parameters. Analyses were conducted using jamovi 1.6 and R 4.1.0 (RStudio, Boston, CA, USA), and R packages RHRV [29,30,31,32].

## 3. Results

### 3.1. Participants 

In our study, twenty-one females and nine males participated and their mean age was 53.8 ± 9.1 years.

### 3.2. Body Mass Composition

The median weight was 87.6 ± 16.9 kg and BMI 32.0 [26.4–34.6]. We found a significant decrease in weight (*p* = 0.022) and a mass of body fat (*p* < 0.001). Furthermore, visceral fat level, visceral area, waist to hip ratio, and abdominal circumference decreased (*p* < 0.001). We obtained the same result in the mass of body fat measurement in the left (*p* = 0.008) and right arms (*p* = 0.005), and in the legs (*p* < 0.001) (Table 1).

We did not find significant changes in other body mass composition parameters such as lean body mass, soft lean mass, skeletal muscle mass, body minerals and proteins, total body fat, and mass of soft lean mass in limbs. 

### 3.3. Blood Tests

During the study, a statistically significant change in glycated hemoglobin levels, cholesterol, and triglycerides levels was not observed (Table 2).

### 3.4. Physical Activity Evaluation Using Fitbit Inspire Physical Activity Tracker Data

Comparing the means of steps during the first and the last months of the trials, the mean of steps decreased by 1870 per day (*p* < 0.001), and the distance decreased by 1.29 km/day. Additionally, we have seen a statistically significant decrease in light activity by 47 min/day (*p* < 0.001), but we did not find any significant change in activity minutes for fairly active and very active physical activity. Sedentary time increased 67.3 min/day, but the change was not statistically significant. The main data is presented in Table 3.

As the raw physical activity data shows, how long the participants were physically active and the impact of physical activity variability on health results were evaluated. 

### 3.5. Physical Activity Variability Evaluation Using Fitbit Inspire Data

For physical activity variability evaluation, we used the Poincaré plot method. As shown in Figure 1, we have data from two different subjects. In the upper charts, we have the first subject, whose SD1 and SD2 (SD12) ratio is close to 1, which means physical activity variability is always high. The second, lower example shows a much lower SD12 ratio, and the short-term variability is much lower than the long-term.

We observed significant positive correlations between changes in lipid values and variability of steps count, distance, and minutes of very active physical activity. It shows a decrease in lipid levels with lower long-term variability and area of an ellipse, but not short-term variability (Figure 2).

The SD1/SD2 ratio showed an opposite relationship with body fat measures indicating that possibly higher short-term variability could be associated with positive changes (Figure 3).

## 4. Discussion

### 4.1. Principal Result

Studies show that the use of consumer-level physical activity trackers could be the alternative to more conservative methods. It has its strengths such as activity trackers that can track PA all the time, activity is leveled in different intensities, it does not require time to fill the forms, and we should not be afraid of retrospective inaccuracy as physical activity can be tracked for an unlimited time. As we found in our study, Mickael Ringeval et al. also found that Fitbit devices in interventions can promote healthy lifestyles in terms of physical activity and weight [33].

The evaluation of physical activity distribution in time is an important factor in assessing a patient’s lifestyle. Constantly higher physical activity has a higher impact on the metabolic effect and body mass composition. Silva BGC et al. research shows that adipose was lower among those who were constantly active or became active [34]. In addition, the study of Carnero EA et al. conducted research with non-diabetic patients for whom bariatric surgery was performed. Physical activity intervention helped patients in the highest quartile of step/day change lose more fat mass, abdominal adipose tissue, and gain skeletal muscle mass [35]. However, we need further research to establish publicly accessible tools, to evaluate physical activity variability, and keep in touch with a patient. Therefore, our research looks for consumer tools and mathematical models to make it adaptable in clinical practice and everyday life.

Our results show significant monthly steps and physical activity decline during the period of the study. This could be referred to the time of the year, and the epidemiologic situation. The study took place from the end of the summer (August) to the middle of wintertime (February), and, usually, people are less physically active at this time of the year. Second, the study started before the second wave of COVID-19 pandemic and ended during the national lockdown. When comparing these two periods of time, before and after lockdown, the steps per day mean decreased by 1852 steps, and on the other hand, the sedentary time increased by 75 min per day (*p* < 0.001).

The change in physical activity in time is one of the most important factors that is hard to assess by questionnaires, but now we can track it with smart bracelets, activity trackers, and cell phones. There was little research on physical activity variability, but there are methods to assess biological signals in other fields such as heart rate, muscle strength, and others. The Poincaré model lets us evaluate and visualize physical activity variability. We discovered a significant correlation between activity intensity variability and weight, but further research with bigger samples and different time intervals should be done. Furthermore, there are other models to evaluate variability, such as a change of coefficient of variant, attractors, two-stage modeling, and data processing with ensemble clustering binning, but the best way is still to conduct future research. 

To use these methods further, we need to test, adapt, and improve every possible data processing method and make it accessible to health specialists for practical purposes and further clinical research. 

### 4.2. Limitations

Our study has limitations. First, it was conducted with a small sample of patients with no control group, and therefore, these approaches should be tested in other settings. Second, it was run over six months, and more extended studies are needed to determine if these shorter-term changes in body weight lead to longer-term changes in preventing diabetes. 

Because of the new strict general data protection regulations in Lithuania, we could not remotely collect patient data for our study. We also could not collect data in between the first and the last visits, because of the strict patient movement restrictions due to the spread of the COVID-19 infection. Therefore, it was not possible to collect data on the actual wear time.

## 5. Conclusions

Data from Fitbit activity trackers create an excellent opportunity to evaluate physical activity for patients at an increased risk of T2D. When assessing physical activity data, doctors should evaluate not just the totals or the medians of the steps count but also physical activity variability in time. The study shows that body fat composition, weight, and total cholesterol may be better linked to the physical activity variability than the total count of physical activity during the research period. 

## Figures and Tables

**Figure 1 ijerph-19-08251-f001:**
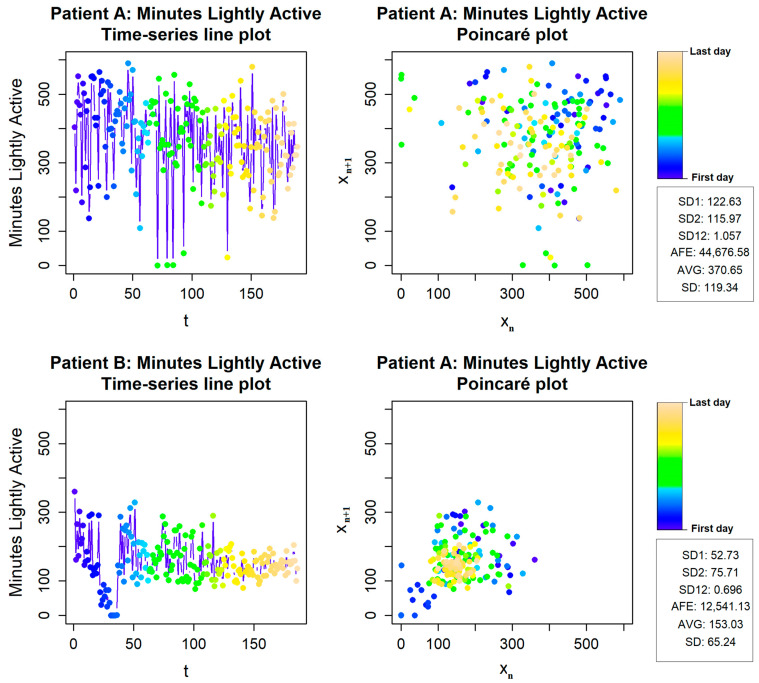
Physical activity variability of patients A and B.

**Figure 2 ijerph-19-08251-f002:**
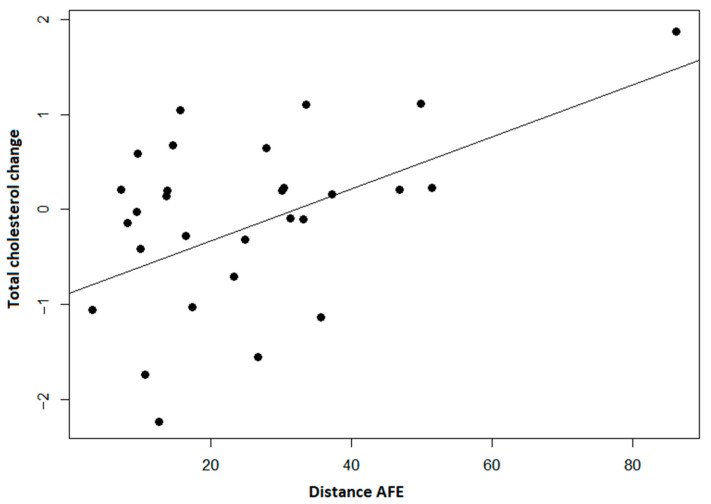
Distance per day area of the fitting ellipse (AFE) and change in total cholesterol shows us that when the fitting ellipse increases, the level of total cholesterol increases (*p* < 0.001).

**Figure 3 ijerph-19-08251-f003:**
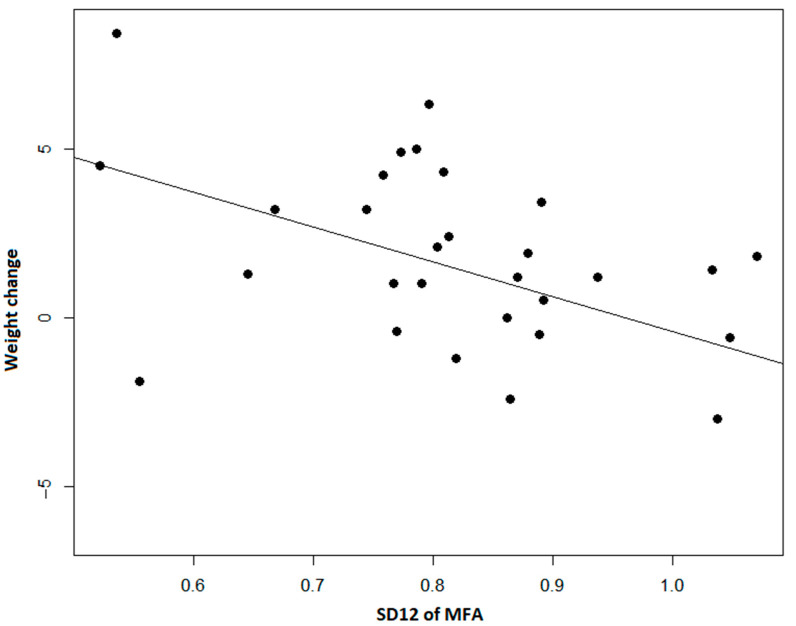
The ratio of SD1 and SD2 of Minutes Fairly Active (MFA) shows us a correlation between lower short-term and long-term variability ratios and weight. The smaller the short-term and long-term variability ratio, the smaller the decrease in weight observed (*p* < 0.001).

**Table 1 ijerph-19-08251-t001:** Body mass composition.

	Baseline (*n* = 30)	Follow-Up—Baseline (*n* = 30)	*p* Value	Effect Size	SRM
Weight [kg] ^#^	87.6 ± 16.9	−1.46 ± 3.31	0.022 *	−0.4406	−0.4411
MBF [kg] ^#^	32.0 ± 9.72	−1.8 ± 2.36	<0.001 *	−0.7617	−0.7627
PBF [%] ^#^	36.1 ± 6.86	−1.43 ± 1.59	<0.001 *	−0.8971	−0.8994
VFL [Units] ^$^	16.0 [13.0–18.0]	−0.867 [0–1.75]	<0.001 *	−0.87619	−0.8337
VFA [cm^2^] ^#^	169 ± 79.1	−21.9 ± 23.9	<0.001 *	−0.9162	−0.9163
WHR ^#^	0.948 ± 0.0912	−0.022 ± 0.0282	<0.001 *	−0.7798	−0.7801
AC [cm] ^#^	98.1 ± 13.0	−2.16 ± 2.84	<0.001 *	−0.7618	−0.7606
MBF Lt.ARM [kg] ^#^	1.93 ± 0.612	−0.0777 ± 0.149	0.008 *	−0.5227	−0.5215
MBF Rt.ARM [kg] ^#^	1.91 ± 0.625	−0.088 ± 0.159	0.005 *	−0.5518	−0.5535
MBF Lt.LEG [kg] ^#^	5.84 ± 1.75	−0.345 ± 0.442	<0.001 *	−0.7810	−0.7805
MBF Rt.LEG [kg] ^#^	5.83 ± 1.76	−0.362 ± 0.444	<0.001 *	−0.8146	−0.8153
MBF Trunk [kg] ^#^	16.5 ± 4.98	−0.924 ± 1.2	<0.001 *	−0.7666	−0.7700

Values are presented as mean ± SD or median [Q1–Q3]. * Two-sided *p* value < 0.05. ^#^ Paired *t*-test, Cohen’s d (effect size). ^$^ Wilcoxon’s matched pairs signed rank test, Rank biserial correlation (effect size). SRM—standardised response mean.

**Table 2 ijerph-19-08251-t002:** Blood tests results.

	Baseline (*n* = 30)	Follow-Up—Baseline (*n* = 30)	*p* Value	Effect Size	SRM
HgbA1c [%] ^#^	5.61 ± 0.352	0.0467 ± 0.2788	0.367	0.1674	0.1675
HgbA1c [mmol/L] ^#^	37.7 ± 3.74	0.5333 ± 2.9564	0.331	0.1804	0.1804
TC [mmol/L] ^#^	5.70 ± 1.14	0.199 ± 1.1188	0.338	0.1779	0.1779
TG [mmol/L] ^$^	1.81 [1.19–2.50]	−0.182 [−0.77–0.303]	0.271	−0.2344	−0.1844
HDL [mmol/L] ^$^	1.29 [1.12–1.52]	0.0287 [−0.1–0.13]	0.593	0.1140	0.1643
LDL [mmol/L] ^#^	3.44 ± 1.04	0.228 ± 1.0574	0.247	0.2156	0.2156

Values are presented as mean ± SD or median [Q1–Q3]. ^#^ Paired *t*-test, Cohen’s d (effect size). ^$^ Wilcoxon’s matched pairs signed rank test, Rank biserial correlation (effect size). SRM—standardised response mean.

**Table 3 ijerph-19-08251-t003:** Data from Fitbit Inspire physical activity tracker.

**Steps per Day**			
Month	Mean of count	Std. Deviation	Coefficient of variance
1st	9908	3625	0.43
2nd	9816	3834	0.41
3rd	8881	3969	0.46
4th	8217	4117	0.53
5th	7971	3597	0.52
6th	8038	3897	0.54
**Activity: Sedentary**			
Month	Mean of minutes	Std. Deviation	Coefficient of variance
1st	804	274	0.24
2nd	805	287	0.19
3rd	816	275	0.18
4th	898	322	0.18
5th	927	290	0.17
6th	871	266	0.25
**Activity: Lightly active**			
Month	Mean of minutes	Std. Deviation	Coefficient of variance
1st	281	117	0.4
2nd	270	116	0.29
3rd	263	121	0.36
4th	216	132	0.54
5th	211	105	0.55
6th	233	99	0.45
**Activity: Fairly active**			
Month	Mean of minutes	Std. Deviation	Coefficient of variance
1st	17	15	1.05
2nd	16	14	1.17
3rd	14	11	1.06
4th	10	10	1.25
5th	11	10	1.49
6th	14	13	1.34
**Activity: Very active**			
Month	Mean of minutes	Std. Deviation	Coefficient of variance
1st	8	8	1.28
2nd	10	10	1.33
3rd	9	10	1.25
4th	10	15	1.43
5th	9	13	1.74
6th	11	14	1.52

## Data Availability

Not applicable.

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
