# Peer review of "Physical Activity Evaluation Using Activity Trackers for Type 2 Diabetes Prevention in Patients with Prediabetes"

_ijerph, 2022, doi:10.3390/ijerph19148251_

Round 1

Reviewer 1 Report

It should be explained more specifically what Fitbit activity trackers are. The data of the participants is insufficient, they should provide more data in this section.

Author Response

Dear Reviewer,

We are grateful for Your notes and observations. We edited our manuscript to Your recommendations.

We specified what Fitbit activity trackers are and added the following paragraph.

2.2. Intervention tools

Every participant was given a Fitbit Inspire activity tracker. Fibit is a consumer product to monitor physical activity like steps, distance, physical activity levels, and sedentary time. These are commonly small devices and are worn on the wrist and can be used individually by the screen on the device or connected by smartphone and mobile application. Fitbit activity tracker uses a microelectronic triaxial accelerometer to capture body motion in 3 dimensions, and then these motions are analyzed by the alghorytmes and converted to the physical activity data. These consumer devices showed respective data accuracy and are more commonly used in clinical trials. [1] Every participant was taught how to use these tools and was given full technical support during the study.

  1. Feehan LM, Geldman J, Sayre EC, Park C, Ezzat AM, Young Yoo J, et al. Accuracy of Fitbit Devices: Systematic Review and Narrative Syntheses of Quantitative Data. JMIR Mhealth Uhealth 2018;6(8):e10527 https://mhealth.jmir.org/2018/8/e10527 [Internet]. 2018 Aug 9 [cited 2022 Jun 20];6(8):e10527. Available from: https://mhealth.jmir.org/2018/8/e10527

As the date was not sufficient we added the age of the patients recruited for the trials.

Participants were selected from 18 to 65 years old, had increased glucose level (5.6 to 6.9 mmol/l), and had no health issues associated with the skeletal or muscular system, or psychical disorders.

Hope we filled Your notes.

Sincerely

Reviewer 2 Report

Methodoloy - I disagree this is a cohort observation study, in a standard cohort study you just follow the population and record them without any interference. This study is a form of intervention study even though there is no control group. You have a pre and post data collection which was the basis of your analysis and results.

Results - Table 1 - what are MBF, PBF, VFL, VFA, WHR, etc, please include a legend to describe them. 

Discussion - Table 3 of the Results on Steps per day - please discuss why step-count declined gradually from the first month (9908) to the fifth month (7971).

Author Response

Dear Reviewer,

We are grateful for Your notes and observations. We edited our manuscript to Your recommendations.

Methodology. We think that You are right and an interventional study without a control group design is more accurate. We based our study on a cohort observational study because we decided to use an activity tracker for all the participants, creating an equal environment, but You gave the right point that however, it is still interference.

We added a legend at the end of the manuscript with all the descriptions.

To explain Your note about step-count decline, we added a paragraph in the discussion section:

Our results show a significant monthly step decline during the period of the study. This could be referred to the time of the year, and the epidemiologic situation. The study took place from the end of the summer (August) to the middle of wintertime (February) and usually, people are less physically active at this time of the year. Second, the study started before the second wave of COVID 19 pandemic and ended during the national lockdown. Comparing these two periods of time, before and after lockdown, the steps per day mean decreased by 1852 steps, and on the other hand, the sedentary time increased by 75 minutes per day (p<0.001).

Hope we filled Your notes.

Sincerely

Reviewer 3 Report

The authors conduct an observational cohort study and aimed to evaluate the use of Fitbit activity trackers in the assessment of physical activity and its variability in time.

There were two visits (thirty volunteers) to the family doctors in Vilnius University Hospital Santaros Klinikos (VUHSK) Family medicine center.

Comments:

Tables 1 and 2

#Paired t-test, Cohen's d (effect size)

$Wilcoxon's matched pairs signed rank test, Rank biserial correlation (effect size)

Tables 1 and 2 should be provided the pre–post changes within groups (mean (standard Deviation, SD) or median (interquartile range, IQR)).

e.g.,

Pre–post changes within groups were estimated via the standardised response mean, with mean differences between T1 and T0 divided by the standard deviation of the difference scores. (Eur Respir J. 2018 Jan 25;51(1). PMID: 29371382)

Author Response

Dear Reviewer,

We are grateful for Your notes and observations. We edited our manuscript to Your recommendations.

We repeated our calculations and added standardized response mean to the table 1 and 2.

Hope we filled Your notes the right way as You recommended.

Sincerely

Reviewer 4 Report

The manuscript of Bliudzius et al presents the results of activity trackers on physical activity levels and body composition in persons with prediabetes. Although the subject is not new the research present adds important information to existing literature on this subject.

While the study seems well designed, there are also some major comments that must be addressed:

1.      Although the English language is good there are some sentences that should be revised as they are difficult to read and have grammar problems

2.      The conclusions of the abstract are not based on the results presented.

3.      Did the authors gather any data on the actual wear time of the devices? This would be important to be presented and discussed as it reflects the adherence.

4.      The main results are the effect of wearable physical activity trackers on the physical activity levels and body compositions. The Discussion should address these results in the context of existing literature. Also, it would be interesting to discuss the importance of objectively assessing the physical activity on the adherence to recommendations.

5.      What would be the explanations for the decrease observed in the physical activity between the first and the last months of the study and possible causes of this decrease?

6.      Also, a discussion on previous data on physical activity variability and what is its impact/mechanisms involved in its metabolic effect should be added.

Author Response

Dear Reviewer,

We are grateful for Your notes and observations. We edited our manuscript to Your recommendations.

We edited our conclusions of the abstract and now they fit the results.

We couldn’t gather any data on the actual wear. We added the paragraph in the limitations section to cover this note.

Because of the new strict general data protection regulations in Lithuania, we couldn’t remotely collect patient data for our study. We also couldn’t collect data between the first and the last visits, because of the strict patient movement restrictions to the spread of COVID 19 infection. So it wasn’t possible to collect data on the actual wear time.

As You noted the Discussion should address these results in the context of existing literature.

We haven’t found a statistically significant association between the means of physical activity levels and body composition due to the small sample size. Also, our study has limitations and lacks a control group to evaluate the impact of physical activity tracker use on physical activity levels, but it gives us a huge knowledge for future research. The existing literature in the field of physical activity variability assessment is limited. So, we are working on a new study design with large sample size, control group, and remote data gathering to answer Your raised question.

To cover your notice about the decrease in physical activity, we added a paragraph in the discussion section.

Our results show a significant monthly step decline during the period of the study. This could be referred to the time of the year, and the epidemiologic situation. The study took place from the end of the summer (August) to the middle of wintertime (February) and usually, people are less physically active at this time of the year. Second, the study started before the second wave of COVID 19 pandemic and ended during the national lockdown. Comparing these two periods of time, before and after lockdown, the steps per day mean decreased by 1852 steps, and on the other hand, the sedentary time increased by 75 minutes per day (p<0.001).

Also, we added a short discussion about previous data on physical activity variability as You suggested.

The evaluation of physical activity distribution in time is an important factor in assessing a patient lifestyle. Constantly higher physical activity has a higher impact on the metabolic effect and body mass composition. Silva BCG et. al. research shows that adipose was lower among those, who were constantly active, or who became active. [36] Also, the study of Carnero EA et. al. made research with non-diabetic patients for whom bariatric surgery was performed. Physical activity intervention helped patients in the highest quartile of step/day change lose more fat mass, abdominal adipose tissue, and gained skeletal muscle mass.[37] But we need further research to establish publicly accessible tools to evaluate physical activity variability and keep in touch with a patient. So our research is looking for the consumer's tools and mathematical model to make it adaptable in clinical practice and everyday life.

Hope we filled Your notes.

Sincerely

Round 2

Reviewer 3 Report

Authors added standardized response mean to the table 1 and 2.

In general, effect size in Tables 1 and 2 should be similar/equal the values of standardised response mean (SRM) when paired t test is used.

Suggestion:

Authors should be provided the values of change Follow-up – Baseline, such as this is paper (Table 4, Eur Respir J. 2018 Jan 25;51(1). PMID: 29371382), and recheck the results of effect size and SRM (some of similar/equal, some of unequal).

It will be a question for this manuscript.

Author Response

Dear Reviewer, 

Thank You for Your comments. 

We repeated the calculations of standardised response mean a few times and we have found the a "bug" in our R+ code. We very appreciate Your note. 

We also provided the values of change Follow-up – Baseline, as suggested in example manuscript. 

Sincerely

Reviewer 4 Report

Thank you for addressing previous comments. I do not have any additional comments at this review round.

Author Response

Dear Reviewer, 

Thank You for Your comments. 

Sincerely.